# YAP Promotes Cell Proliferation and Stemness Maintenance of Porcine Muscle Stem Cells under High-Density Condition

**DOI:** 10.3390/cells10113069

**Published:** 2021-11-08

**Authors:** Zheng Liu, Ling Lin, Haozhe Zhu, Zhongyuan Wu, Xi Ding, Rongrong Hu, Yichen Jiang, Changbo Tang, Shijie Ding, Renpeng Guo

**Affiliations:** 1College of Food Science and Technology, Nanjing Agricultural University, Nanjing 210095, China; 2019108079@njau.edu.cn (Z.L.); 2017108080@njau.edu.cn (L.L.); 2016108086@njau.edu.cn (H.Z.); 2021208035@stu.njau.edu.cn (Z.W.); 2019208007@njau.edu.cn (X.D.); 2019808141@njau.edu.cn (R.H.); 2020208036@stu.njau.edu.cn (Y.J.); tangcb@njau.edu.cn (C.T.); 2National Center of Meat Quality and Safety Control, Key Laboratory of Meat Processing and Quality Control, Key Laboratory of Meat Processing, Nanjing Agricultural University, Nanjing 210095, China

**Keywords:** muscle stem cells, high-density culture, Hippo-YAP, cell proliferation, stemness maintenance, cultured meat

## Abstract

Muscle stem cells (MuSCs) isolated ex vivo are essential original cells to produce cultured meat. Currently, one of the main obstacles for cultured meat production derives from the limited capacity of large-scale amplification of MuSCs, especially under high-density culture condition. Here, we show that at higher cell densities, proliferation and differentiation capacities of porcine MuSCs are impaired. We investigate the roles of Hippo-YAP signaling, which is important regulators in response to cell contact inhibition. Interestingly, abundant but not functional YAP proteins are accumulated in MuSCs seeded at high density. When treated with lysophosphatidic acid (LPA), the activator of YAP, porcine MuSCs exhibit increased proliferation and elevated differentiation potential compared with control cells. Moreover, constitutively active YAP with deactivated phosphorylation sites, but not intact YAP, promotes cell proliferation and stemness maintenance of MuSCs. Together, we reveal a potential molecular target that enables massive MuSCs expansion for large-scale cultured meat production under high-density condition.

## 1. Introduction

Meat is a good source of high quality proteins. To satisfy the world’s growing population and increasing demands for food of animal origin, the meat industry would have to increase production by approximately 50–100% in the next three decades [1]. Yet, the traditional meat production is currently facing tremendous challenges, including overuse of arable lands and water, animal-borne pathogens, environmental footprints, and poor animal welfare conditions [2]. Thus, it is necessary to develop new technologies for sustainable meat production and supply.

Cultured meat, also termed as cultivated meat, cell-based meat, or clean meat, is a promising alternative for future meat consumption [3,4]. Lab-based cell cultured meat production is achieved through isolation and expansion of animal stem cells, committed myogenic or adipogenic differentiation, and food processing [3]. In contrast to traditional meat production, cultured meat technology exhibits distinct advantages in sustainability, animal welfare and public health [4]. Inspiringly, cultured meat has been tasted in the UK, China and other parts of the world. For instance, in 2013, the first cell-cultured hamburger, which was developed by Dr. Mark Post at the University of Maastricht, was cooked and tasted in London, England [5]. In China, cultured meat was firstly tasted in Nanjing Agricultural University in 2020, which was pioneered by Dr. Guanghong Zhou.

Myogenic stem cells, such as muscle stem cells (MuSCs) or satellite cells, are basically used for cultured meat research [6]. In the muscle tissues, quiescent MuSCs are found between the basement membrane and sarcolemma of muscle fibers [7]. During development or upon injury, these Pax7/Pax3-expressing MuSCs are activated to proliferate, and terminally form differentiated multinucleated myofibers [8], accompanied by staged expression of myogenic factors, including Myf5, MyoD, Mrf4, Myogenin (MyoG) [9]. Once isolated and cultured in controlled growth conditions, MuSCs can either replicate to increase the number of cells or differentiate into multinucleated myotubes to produce cell-based meat [6]. Currently, one of the main obstacles for cultured meat production derives from the limited capacity of large-scale amplification of MuSCs [10]. The existence of contact inhibition under high-density culture condition restricts clinical or industrial-scale amplification of stem cells [11,12]. Investigations should be performed to elucidate the proliferation rate, stemness maintenance and underlying signaling transductions of MuSCs under high-density condition.

Hippo-Yap pathway has been recognized by numerous studies to be an important pathway controlling cell contact inhibition [13,14,15]. The Hippo pathway is highly evolutionarily conserved in mammals, which mainly comprises kinase cascade that primarily target paralogous transcriptional regulators YAP and TAZ [16,17,18]. Activation of the Hippo pathway by upstream signals results in the inactivation of YAP by LATS1/2-mediated phosphorylation in the HxRxxS motif. Phosphorylated YAP is sequestered in the cytoplasm and degraded by ubiquitin-proteasome system (UPS). Conversely, when the Hippo pathway is inactivated, unphosphorylated YAP translocates into the nucleus where it associates with TEAD family transcription factors to regulate gene expression and promote cell proliferation [19,20]. The Hippo pathway has been demonstrated as the main locus where various pathways sensing cell density are integrated to mediate cell growth [21,22]. Further, YAP signaling is implicated in regulation of murine myogenic fate decisions [23]. In domestic animal-derived MuSCs, it is unclear whether or by what mechanism cell density regulates YAP and the maintenance of stemness.

In this study, we find that proliferation and differentiation capacity of porcine MuSCs are impaired when cultured under high-density condition, indicating the existence of cell contact inhibition. High-density cell culture leads to increased phosphorylation level of YAP. When treated with lysophosphatidic acid (LPA), the activator of YAP, porcine MuSCs exhibit increased proliferation and elevated expression of myogenic factors, and up-regulated differentiation ability in comparison with control cells. Finally, constitutively active porcine YAP (serine to alanine mutations in the HxRxxS motifs), but not wide type YAP, activates cell growth and expression of genes contributing to stemness maintenance. Our results demonstrate a pivotal role of YAP in cell density-mediated proliferation and fate determinations of porcine MuSCs.

## 2. Materials and Methods

### 2.1. Ethical Statement

The porcine muscle tissues used in this study were derived from a 1 week old pig, which was approved by the Animal Ethics Committee, Nanjing Agricultural University, China (Approval Code: IACVC2020172; Approval Date: 10 November 2020).

### 2.2. Pig MuSCs Isolation

Pig MuSCs isolation was performed as described previously [24]. Briefly, approximate 0.25 kg thigh and hip fresh muscle tissue was dissociated with collagenase D (2 μg/mL, Roche, Mannheim, Germany, Cat# 11088866001) and dispase II (2 mg/mL, Roche, Cat# 4942078001) in DMEM (Invitrogen, Bleiswijk, The Netherlands, Cat# c11330500bt) supplement with 3% penicillin-streptomycin (Gibco, 15140122, Carlsbad, CA, USA, Cat# 15140122). Collagen and other impurities are removed by cell strainer. The cells were reconstituted with 1% BSA in PBS and stained with an antibody cocktail consisting APC-conjugated anti-pig CD31 (1:20, BIO-RAD, Richmond, CA, USA, Cat# MCA1746APC), Alexa Fluor 647 anti-pig CD45 (1:20, BIO-RAD, Richmond, CA, USA, Cat# MCA1222A647), PE-conjugated anti-human CD56 (1: 40, BioLegend, San Diego, CA, USA, Cat# 304606), and Alexa Fluor 488 anti-human CD29 (1:40, BioLegend, San Diego, CA, USA, Cat# 303016). Cell sorting was performed with a BD Influx cell sorter using 488, 561, and 640 nm lasers. The viable CD31^−^CD45^−^CD56^+^CD29^+^ cells were isolated.

### 2.3. MuSCs Culture and Differentiation

Dishes (Corning, NY, USA, Cat# 430167) were coated with 0.05% rat tail collagen type I (Corning, Cat# 354236). FACS isolated pig MuSCs were cultured on collagen-coated dishes in F10 medium (Gibco, Carlsbad, CA, USA, Cat# 11550-043) containing 20% fetal bovine serum (Gibco, Cat# 10270-106), 5 ng/mL bFGF (R&D, Carlsbad, CA, USA, Cat# 233-FB-500/CF) and 1% penicillin-streptomycin. For serial expansion, cells were passaged to maintain a density of <60% confluence and counted at each passage [25]. The bright field images were acquired by Leica microsystem (DMI600B). For cell differentiation, pig MuSCs were plated onto 2% matrigel (Corning, Cat# 356234)-coated dishes in F10 medium containing 20% fetal bovine serum, 5 ng/mL bFGF and 1% Penicillin-Streptomycin at density of 1.25 × 10^5^/cm^2^. The next day, medium was changed to DMEM (Gibco, Shanghai, China, Cat# C11995500BT) with 2% horse serum (Hyclone, Logan, UT, USA, Cat# SH30074.02) and 1% penicillin-streptomycin. Cell differentiation was induced for 5 days and committed to further experiments.

### 2.4. Clone of the Full-Length Pig YAP1 CDS

Total RNA was extracted from pig MuSCs. PCR primers flanking pig *YAP1* were designed based on the published predicted sequence in NCBI database (Accession#: XM_021062706). PCR experiments were performed with 2 × Phanta^®^ Max Master Mix (Dye Plus) (Vazyme, Nanjing, China, Cat# P525-01) following the manufacturer’s protocol. We then cloned the resultant *YAP1* CDS into the pEASY^®^-Blunt Simple Cloning Vector (Transgen, Nanjing, China, Cat# CB11) and verified the sequence via Sanger sequencing. The *YAP1* CDS sequences and protein sequences were aligned by Clustal Omega (https://www.ebi.ac.uk/Tools/msa/clustalo/, 29 October 2021) supported by The European Bioinformatics Institute (EBL). The primers used were (accggt indicate restriction site for AgeI): F: 5′-accggtATGGATCCCGGGCAGCAGCAGCCGC-3′, R: 5′-accggtCTATAACCATGTAAGAAAGCTTTC-3′.

### 2.5. Cell Transfection and Retroviral Infection

To generate wild-type or mutant YAP-expressing cells, lentiviral constructs with doxycycline (Dox)-inducible system were established. Pig *YAP* CDS with all five phosphorylation sites mutation (serine to alanine, 5A) was obtained with Fast MultiSite Mutagenesis System (Transgen, Cat# FM201) following the manufacturer’s protocol. Lentiviral plasmid pLVX-TetOne-Puro (Wuhan Miaolingbio, Cat# P1686) contains tetracycline responsive promoter and modified reverse tetracycline-controlled transactivator (rtTA), which responds strongly to Dox induction. Wild-type and mutant *YAP* CDS were cloned into pLVX-TetOne-Puro. The above recombinant plasmids, together with auxiliary plasmids pMD2.G (Addgen# 12259, Watertown, MA, USA) and psPAX2 (Addgen# 12260, Watertown, MA, USA), were transfected into 293T cells (Procell, Wuhan, China, Cat# CL-0005) using Lipofectamine^®^ 3000 (Invitrogen, Carlsbad, CA, USA, Cat# L3000008) according to the manufacturer’s instructions. Then, 24 h post transfection, lentiviral supernatant was supplemented with 4 μg/mL polybrene (Santa Cruz, Dallas, TX, USA, Cat# 000825), filtered through 0.45 μm filters (MerckMillopore, Billerica MA USA, Cat# SLHV033RB), and used to infect MuSCs. Then, 12 h after infection, virus-containing supernatant was removed and changed to fresh medium. Then, 24 h later, cells were selected with 2 μg/mL puromycin (Beyotime, Nanjing, China, Cat# ST551) for 5 days. Then, survival cells were gradually diluted to obtain single cell-derived clones. 2 μg/mL Dox (Beyotime, Nanjing, China, Cat# ST039A) was used to induce *YAP* expression and for further experiments.

### 2.6. Gene Expression Analysis by Quantitative RT-PCR

Total RNA was isolated from MuSCs using Trizol reagent (Invitrogen, Cat# 15596018). cDNA was synthesized by reverse transcription using PrimerScript RT Master Mix (TAKARA, Cat# RR036A). Real-time quantitative PCR reactions were set up in triplicate with the TB Green^®^ Premix Ex Taq^TM^ II (TAKARA, Nanjing, China, Cat# RR820) and run on the QuantStudio 6 Flex (Thermo, QuantStudio 6Flex). Glyceraldehyde 3-phosphate dehydrogenase (GAPDH) was used as an internal control. The primers are listed in Table 1.

### 2.7. Immunofluorescence Staining

For immunofluorescence staining, MuSCs were cultured on Nunc Glass Base Dish (Thermo, Cat# 150682) and fixed with 4% paraformaldehyde (PFA) (Beyotime, Nanjing, China, Cat# P0099) overnight at 4 °C, then permeabilized with 0.5% Triton X-100 in PBS for 30 min, washed three times with PBS. Samples were incubated overnight at 4 °C with primary antibodies: YAP (1:800, ABclonal, Wuhan, China, CAT#A1002), MYHC (1:500, Abcam, Cambridge, UK, CAT# ab37484), then washed three times, incubated with secondary antibodies: Alexa Fluor 594 goat anti-mouse IgG(H+L) (Invitrogen, Cat# A11005) or Alexa Fluor 488 goat anti-Rabbit IgG(H+L) (Invitrogen, Carlsbad, CA, USA, Cat# A11034) (1:500) for 1 h and then washed three times. Samples were mounted with VECTASHIELD mounting medium with DAPI (Vector Laboratories, Burlingame, CA, USA, Cat# H-1200). Images were obtained using a laser scanning confocal microscope (Leica, TCS SP8 X, Watzlar, Germany).

### 2.8. Western Blot

The cells were collected and lysated with RIPA buffer (Byotime, Cat# P0013B) complemented with PMSF (Beyotime, Cat# ST506), protease inhibitor (Beyotime, Cat# P1050), and phosphatase inhibitors (Beyotime, Cat# P1096). Protein concentrations were determined using BCA protein assay kit (Thermo, Cat# 23225). SDS-PAGE electrophoresis was carried out in 4–20% precast polyacrylamide gels (Genscript, Nanjing, China, Cat# M00625) and blotted onto polyvinylidene difluoride (PVDF) membranes. Membranes were blocked for 1 h with 5% skimmed milk powder in TBST and probed overnight at 4 °C with primary antibodies: phosphorylated-YAP (1:1000, Danvers, MA, USA, Cell Signaling, CAT#13008), YAP (1:2000, ABclonal, CAT#A1002), MYHC (1:1000, Abcam, CAT# ab37484), GAPDH (1:2000, Millipore, Darmstadt, Germany, CAT# MAB374). Secondary antibodies HRP conjugated goat anti-Rabbit IgG (Cwbiotech, Nanjing, China, Cat# CW0103S) or goat anti-mouse IgG (Cwbiotech, Cat# CW2333S) were diluted at 1:2000. Protein bands were visualized using SuperSignal^TM^ West Pico Chemiluminescent Substrate (Thermo, Carlsbad, CA, USA, Cat# 34580) under ImageQuant 4000 (General Electric, Boston, NY, USA). For protein quantification, Quantity One was used to analyze grayscale values.

### 2.9. Statistical Analysis

Data are presented as mean ± SD and were analyzed using Graphpad Prism V9. For comparisons of two groups, a two-tailed Students’ *t*-test was used. For comparisons of three or more than three groups, a one-way ANOVA with Fisher’s protected least significant difference (PLSD) or two-way ANOVA with Sidak test was used. Statistically significance was defined as *p* < 0.05, *p* < 0.01, *p* < 0.001 or *p* < 0.0001.

## 3. Results

### 3.1. High-Density Cell Culture Impairs Proliferation and Differentiation Capacity of MuSCs

To determine density-mediated regulation of stemness maintenance, MuSCs were seeded at different densities to collagen-coated plates at 2.7 × 10^3^/cm^2^ and 2.2 × 10^4^/cm^2^, which were defined as low-density and high-density culture, respectively (Figure 1A). The confluence of cells was apparently higher under high-density than low-density condition (Figure 1A). After three days post seeding, MuSCs expanded for approximate 6 folds at low density, while only 2 folds at high density (Figure 1B), indicating cell contact inhibition in porcine MuSCs at high density. The mRNA expression level of *PAX7*, the master regulator of MuSCs [26], was significantly up-regulated to ~3.3 folds in MuSCs seeded at high density compared with cells seeded at low density (Figure 1C). Furthermore, expression levels of *MYOD* and *Myogenin* (*MYOG*), two factors characterizing myogenic precursors and myogenic commitment, respectively [27], were also increased to ~4.5 folds and ~9 folds in high-density cultured MuSCs compared with cells under low density condition (Figure 1D,E). Markedly elevated expression of *MYOG* indicated that MuSCs cultured under high density were prone to differentiation. Protein levels of PAX7, MYOD, and MYOG were consistently elevated in MuSCs under high-density condition in comparison with low-density cultured cells (Figure 1G–J). We also detected expression of *Sprouty1 (SPRY1)*, which regulates quiescence state of MuSCs [28], and found that high-density culture significantly up-regulated its mRNA expression level in MuSCs (Figure 1F).

We then assessed the differentiation capacity of MuSCs cultured under low and high-density conditions. In response to differentiation cues, myogenic progenitors become elongated myocytes, then fuse to form multinucleated myotubes characterized by the expression of myosin heavy chain (MYHC) [29] (Figure 2). Low and high-density cultured MuSCs both successfully differentiated into mature myotubes, while high-density cells derived myotubes exhibited significantly decreased mRNA levels (Figure 2A) and similar protein levels of MYHC (Figure 2B,C). Moreover, compared with low-density cultured cells, MuSCs collected from high-density condition yields lower percentage of myotubes with more than three nuclei in total nuclei by ~12% (Figure 2D,E).

Taken together, these results suggest that cell proliferation and differentiation potential are significantly damaged by high-density culture of porcine MuSCs.

### 3.2. More YAP Is Phosphorylated and Located in Cytoplasm upon High-Density Culture

We then examined whether YAP was involved in cell density-mediated stemness regulation in porcine MuSCs. Expression levels of *YAP* measured by quantitative real-time PCR (qPCR) showed no significant differences between MuSCs cultured in low or high densities (Figure 3A). As a transcriptional regulator, YAP functions only when located in the nucleus and in the non-phosphorylated form [30]. Unexpectedly, total protein levels of YAP were increased in MuSCs under high density (Figure 3B,C). Meanwhile, phosphorylated YAP was also up-regulated upon high-density culture (Figure 3B,D). Relative phosphorylation ratio of YAP (p-YAP/total YAP) was significantly elevated by ~2.8 folds in high-density cultured MuSCs in comparison with low density grown-MuSCs (Figure 3E). By immunofluorescence, YAP was located in the nuclei in the majority of MuSCs (~72%) seeded at low density (Figure 3F,G). However, YAP was clearly sequestered in the cytoplasm, in the majority of high-density cultured MuSCs (~80%, Figure 3F,G). Further, the relative fluorescence intensities for nuclear YAP and the nuclear/cytoplasmic fluorescence ratio of YAP were significantly decreased in MuSCs under high-density conditions (Figure 3H,I). These data imply that in porcine MuSCs, high-density culture leads to cytoplasmic localization and elevated phosphorylation levels of YAP, which might hinder the ability of YAP to regulating cell proliferation. In our study, phosphorylated YAP seems not to be degraded in the cytoplasm, indicating defective UPS or autophagy-mediated degradation process in high-density cultivated MuSCs, which requires further investigations (see also discussion).

### 3.3. LPA Promotes Proliferation and Differentiation Potential of MuSCs

LPA was reported as an activator of YAP [31], we tested whether activating YAP by LPA would promote cell proliferation and stemness maintenance. At low cell density, cell proliferation was significantly elevated by 10 μM LPA treatment and further enhanced by 25 μM LPA (Figure 4A). When under high-density condition, only treated with higher concentration of LPA, MuSCs displayed higher proliferation capacity by ~50% (Figure 4B). We then detected expression of genes regulating self-renewal and differentiation of MuSCs by qPCR assay. *PAX7* and *MYOD* were up-regulated upon LPA treatment regardless of cell densities at two concentrations, except *MYOD* in 25 μM LPA-treated MuSCs under low density (Figure 4C,D). Nonetheless, thought with slight fold changes, *MYOG* and another myogenic regulatory gene *Caveolin-3* (*CAV3*) were also increased after addition of LPA (Figure 4C,D).

Then, we examined the effects of LPA on differentiation of MuSCs. Terminal differentiated cells derived from control and LPA-treated MuSCs cultured under high density showed similar mRNA levels of *MYHC* (Figure 5A). Protein levels of MYHC were significantly up-regulated in LPA-treated MuSCs derived cells (Figure 5B,C). Immunofluorescence staining indicated that differentiated myotubes from LPA-treated MuSCs contained more nuclei per cell than control myotubes (Figure 5D,E). Collectively, the activation of YAP by LPA promotes cell proliferation and maintains gene expression regulating self-renewal of MuSCs, and enhances the differentiation potential of MuSCs.

### 3.4. Phosphorylation Sites of YAP Are Highly Conserved among Various Species

To determine the direct association between YAP and cell proliferation and stemness maintenance, we sought to clone the porcine *YAP* CDS. There are only predicted porcine *YAP1* mRNA sequences in NCBI nucleotide database (Accession #: XM_021062706). We designed primers based on the above sequence and successfully obtained the PCR products. By Sanger sequencing, verified full-length porcine *YAP1* CDS was achieved (Appendix A). The confirmed nucleic acid sequence showed 96% identity to previously predicted sequence. (Appendix A). As phosphorylation sites are key regulatory regions directing the stability and activity of YAP protein, we then searched for phosphorylation sites in porcine YAP protein. Interestingly, by protein sequence alignment, we found that all five potential phosphorylation motifs HxRxxS (H: histidine, R: Arginine, S: serine) were highly conserved in pig and other four mammals (Figure 6). Based upon, it is feasible to manipulate the activity of YAP protein by targeting the consensus motifs. Together, for the first time, porcine *YAP* CDS is cloned and verified via sequencing, and porcine YAP protein contains all five potential phosphorylation motifs.

### 3.5. Constitutively Active Porcine YAP Promotes Proliferation and Stemness Maintenance of MuSCs

Next, we tested the effect of YAP signaling on MuSCs through forced expression of *YAP1*. An exogenous porcine *YAP1* gene driven by a doxymycin (Dox)-inducible promoter was integrated into a lentivirus-based plasmid, resulting in a Dox-inducible *YAP1* transgene through virus packaging and integration. Using such a system, we overexpressed wild-type (WT) *YAP* and constitutively active *YAP* with serine to alanine mutation in all five HxRxxS motifs (5A). Upon Dox induction, expression of YAP was ~10-fold higher in WT YAP-treated MuSCs and further ~20-fold in 5A YAP-treated MuSCs compared with cells transfected with empty vector control (Figure 7A). Constitutively active porcine YAP, but not intact YAP, increased cell proliferation of MuSCs under both low and high-density culture conditions (Figure 7B). When cultivated under low condition and treated by Dox, expression levels of *PAX7*, *MYOD,* and *MYOG* did not change regardless of different levels of YAP (Figure 7C–E). In contrast, under high-density condition, 5A YAP-treated MuSCs, but not WT YAP-transfected cells, exhibited significantly up-regulated expression levels of *PAX7* and *SPRY1*, and down-regulated expression levels of *MYOD* and *MYOG* in comparison with control MuSCs (Figure 7C–F). Collectively, these results suggest that constitutively active YAP can promote cell proliferation and stemness maintenance of MuSCs, especially under high-density culture condition. 

## 4. Discussion

We show that cell contact inhibition can be found in ex vivo cultured porcine MuSCs. When cultured under high-density, MuSCs exhibit defective capacities of proliferation and differentiation, limiting large scale production of cultured meat. Then, we focus on Hippo-YAP pathway, which is reported to play central roles in controlling cell contact inhibition [13,14,15]. Unexpectedly, protein levels of total and phosphorylated YAP are both elevated in MuSCs grown in high-density (Figure 3B–D), which is scarcely observed in previous studies [13,15,21,32]. Our data further suggest that more phosphorylated and cytoplasmic YAP proteins are accumulated in high-density cultured MuSCs than low-density grown-MuSCs (Figure 3D–I). These results indicate that upon high-density culture, though total YAP protein levels are up-regulated, most of them are phosphorylated, located out of the nucleus and not functional. Intriguingly, in high-density culture MuSCs, redundant and phosphorylated YAP seems not to be degraded via UPS, as previously reported [18,20]. Our unpublished multi-omics data suggest that a group of genes and proteins associated with E2 ubiquitin-conjugating enzymes and E3 ubiquitin-protein ligases are greatly down-regulated in MuSCs under high-density condition. This indicate the disordered protein degradation system in MuSCs upon high-density culture and may underlie the uncommonly elevated phosphorylated and pan-YAP levels observed herein. Moreover, YAP can also be degraded by autophagy under certain circumstances [33,34]. Thus, it is interesting to examine autophagic flux changes and autophagy-lysosome mediated YAP degradation in high-density cultured MuSCs in consequent studies.

To clarify the role of YAP signaling in porcine MuSCs, we firstly culture cells with YAP activator LPA and find that LPA treatment significantly enhance cell proliferation and differentiation of MuSCs (Figure 4). Given the potential non-specific effects of LPA treatment, we then construct inducible YAP overexpressed MuSCs, and find that only constitutively active YAP with deactivated phosphorylation sites, but not wild-type YAP can promote proliferation and stemness maintenance of porcine MuSCs (Figure 7). These data further demonstrate the requirement of unphosphorylated state and nuclear localization of YAP in cellular regulation [15,21,22]. Our study clarify that Hippo-YAP pathway can be used as a potential target to yield sufficient cells for large-scale production of cultured meat.

Our results demonstrate that expression levels of MYOG, PAX7, and SPRY1 are all up-regulated in MuSCs seeded at high density (Figure 1C–F). Elevated expression level of MYOG in MuSCs indicates that myogenic cells are prone to differentiate upon confluence. It seems contradictory that the expressions of both MYOG and PAX7 are upregulated in MuSC grown under high-density conditions. Interestingly, PAX7 is an essential transcription factor not only in self-renewing but also in quiescent MuSCs [35]. Further, in our analysis, expression of Sprouty1 (SPRY1), an inhibitor of receptor tyrosine kinase signaling, which promotes MuSCs to enter the quiescence state [36,37], is also elevated upon high-density culture (Figure 1F). Thus, our data suggest that under high-density culture, a subpopulation of MuSCs tend to differentiate with elevated expression of MYOG and another distinct portion of cells are committed to the resting, PAX7^+^/SPRY1^+^ state. It is quite interesting to further reveal and modulate the heterogeneity of porcine MuSCs. Moreover, when YAP is constitutively activated in MuSCs, we observed increased expression of *PAX7* and *SPRY1*, together with decreased expression of *MYOD* and *MYOG* (Figure 7C–F). MYOG^−^/MYOD^−^/PAX7^+^ MuSCs are considered to be non-cycling [35], which may explain the significantly but slightly elevated proliferation ratio of YAP-overexpressed MuSCs (Figure 7B). These data indicate that constitutively YAP prevents spontaneous cell differentiation while promotes more MuSCs to enter a quiescent state. It is reasonable to target molecular pathways regulating quiescent states to further promote cell proliferation of porcine MuSCs at high density.

Herein, we strengthen the role of YAP signaling in regulating proliferation and stemness maintenance of MuSCs under high-density conditions. Nonetheless, some other essential pathways may also participate in cell contact-dependent regulation of MuSCs. Recently, Buikema et al., reported that in human induced pluripotent stem cell-derived cardiomyocytes (hiPSC-CMs), which can also be restricted by cell contact inhibition, both LEF/TCF activity triggered by Wnt signaling and AKT phosphorylation, but not YAP signaling are required for persistent hiPSC-CMs proliferation [38]. Additionally, as widely studied in human and murine, the MAPK signaling, Notch signaling, and FGF/IGF signaling are recognized to be important regulators of proliferation and differentiation of MuSCs [8]. Further investigations should be performed to verify the changes and roles of these pathways in cell density-mediated stemness regulation of porcine MuSCs.

The traditional genomic modification methods via exogenous gene integration and expression are used in our study. For complicated reasons, transgenic products are not widely accepted by consumers, and more reliable and acceptable methods of gene editing can be used in cultured meat research. For example, through CRISPR/Cas9-mediated gene knockout, negative regulators upstream of YAP (Mst1/2 or Lats1/2) can be depleted to achieve sustained activation of YAP signaling [39]. Additionally, YAP and other positive regulators of MuSCs, such as Pax7, can be activated endogenously via CRISPR/dCas9 (d: dead) system without introducing exogenous genes [40]. With efficient, precise, and controllable procedures, novel gene editing systems represented by CRISPR can effectively reduce the risk of uncertainty caused by gene manipulation.

In addition to LPA, the conventional activator of YAP, a more specific and potent YAP activator, termed PY-60, has recently been developed. PY-60 is demonstrated to robustly activates YAP transcriptional activity in vitro and in vivo by targeting annexin A2 [19]. PY-60 holds therapeutic potential in pathological conditions aggravated by insufficient cell proliferation and repair. Additionally, it is intriguingly to investigate the effects of PY-60 on cell proliferation and stemness maintenance of MuSCs in future studies.

## 5. Conclusions

In summary, our data reveal the existence of cell contact inhibition in cultured porcine MuSCs. By treatment with LPA and forced expression of *YAP1*, we demonstrated the role of YAP signaling in promoting proliferation and stemness maintenance of porcine MuSCs. Our study indicates a strategy for the massive expansion of porcine MuSCs with YAP activation, which can be applied in large-scale cultured meat production.

## Figures and Tables

**Figure 1 cells-10-03069-f001:**
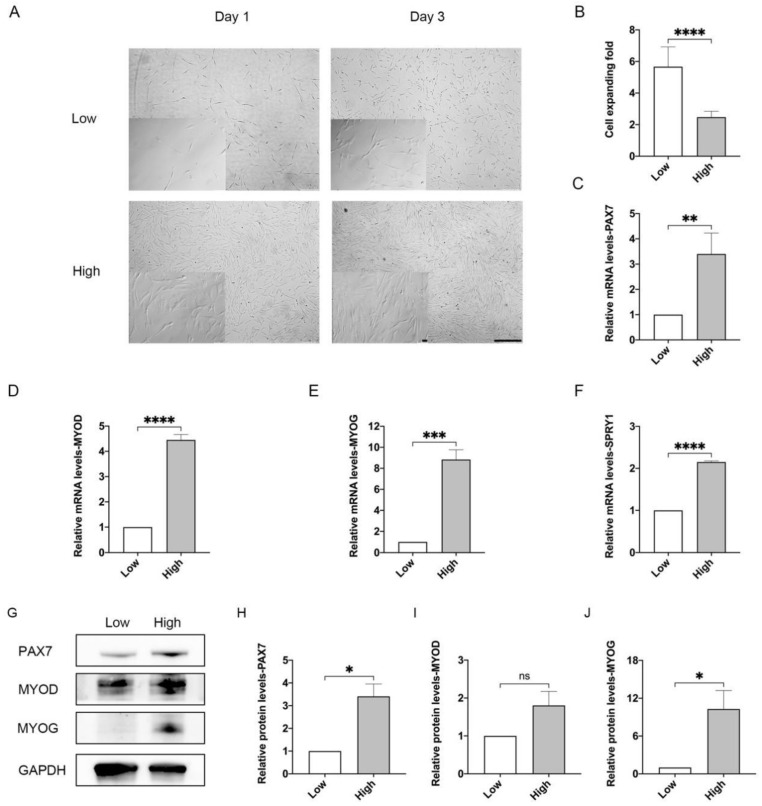
Defective proliferation and differentiation in MuSCs cultured under high-density condition. MuSCs were plated at densities of 2.7 × 10^3^/cm^2^ (low density) and 2.2 × 10^4^/cm^2^ (high density) at day 0. (**A**) Microscopic images of porcine MuSCs cultured in low or high conditions under bright field after one or three days post seeding. Scale bar, 100 μm. (**B**) Cell expanding fold when harvested after three days of culture. (**C**–**F**) Expression by qPCR of *PAX7*, *MYOD*, *MYOG,* and *SPRY1* in MuSCs under low and high-density conditions. (**G**) Western blot analysis of PAX7, MYOD, MYOG in MuSCs. (**H**–**J**) Relative protein levels of PAX7, MYOD, and MYOG normalized to GAPDH indicated in (**G**) by Quantity One software. Mean ± SD from three independent experiments. Significance was analyzed by Students’ *t*-test, * *p* < 0.05, ** *p* < 0.01, *** *p* < 0.001, **** *p* < 0.0001, ns, not significant (*p* > 0.05).

**Figure 2 cells-10-03069-f002:**
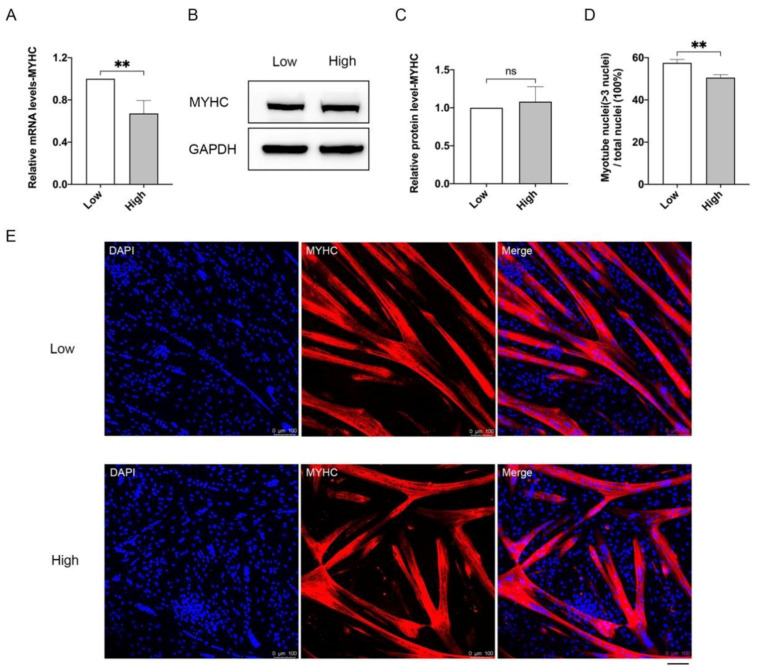
The differentiation capacity of MuSCs under low and high densities. MuSCs were seeded at 2.7 × 10^3^/cm^2^ (low) and 2.2 × 10^4^/cm^2^ (high), cells were harvested after three days culture and seeded onto Matrigel-coated dishes at equal densities to differentiate for 5 days. (**A**) qPCR analysis of *MYHC* mRNA levels. (**B**) MYHC protein levels by Western blot. (**C**) Relative protein levels of MYHC normalized to GAPDH indicated in (**B**) by Quantity One software. (**D**) Quantification of the percentage of myotubes with more than three nuclei. (**E**) Immunofluorescence staining of MYHC (red) and nucleus (blue). Mean ± SD from three independent experiments. Significance was analyzed by Students’ *t*-test. ** *p* < 0.01, ns, not significant (*p* > 0.05).

**Figure 3 cells-10-03069-f003:**
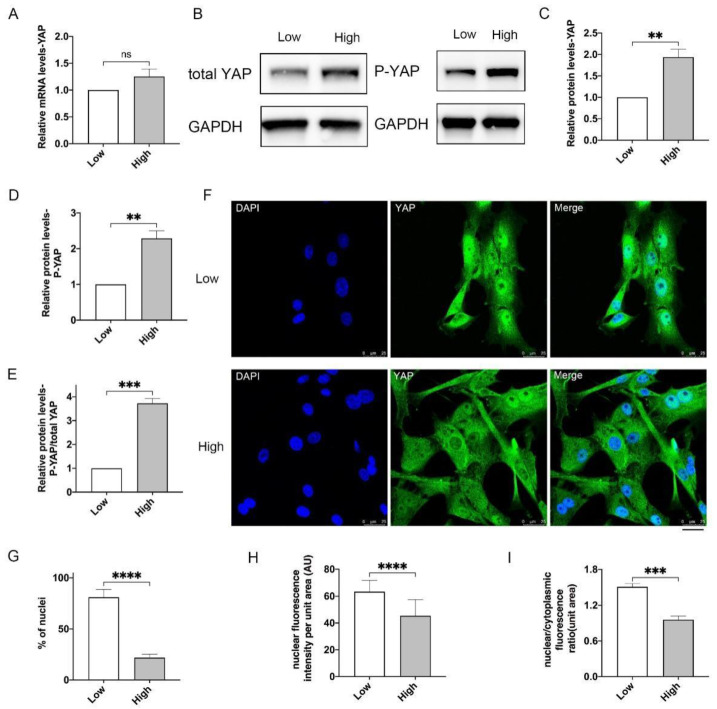
More phosphorylated and cytoplasmic YAP in high-density cultured MuSCs. (**A**) Expression of YAP in MuSCs by qPCR analysis. (**B**) Western blot analysis of total YAP and phosphorylated YAP (P-YAP). (**C**,**D**) Relative preotein levels of YAP and P-YAP normalized to GAPDH indicated in (**B**). (**E**) Quantification of relative p-YAP protein levels to total YAP levels from (**B**). (**F**) Immunofluorescence of YAP (green) showing cytoplasmic and nuclear localization of YAP. Scale bar, 25 μM. (**G**) Percentage of the MuSCs with co-localized YAP and nuclei from (**F**). (**H**) Quantification of nuclear fluorescence intensity per unit area by Image J from (**F**). (**I**) Quantification of nuclear/cytoplasmic fluorescence ratio by Image J from (**F**). Mean ± SD from three independent experiments. Significance was analyzed by Students’ *t*-test. ** *p* < 0.01, *** *p* < 0.001, **** *p* < 0.0001, ns, not significant (*p* > 0.05).

**Figure 4 cells-10-03069-f004:**
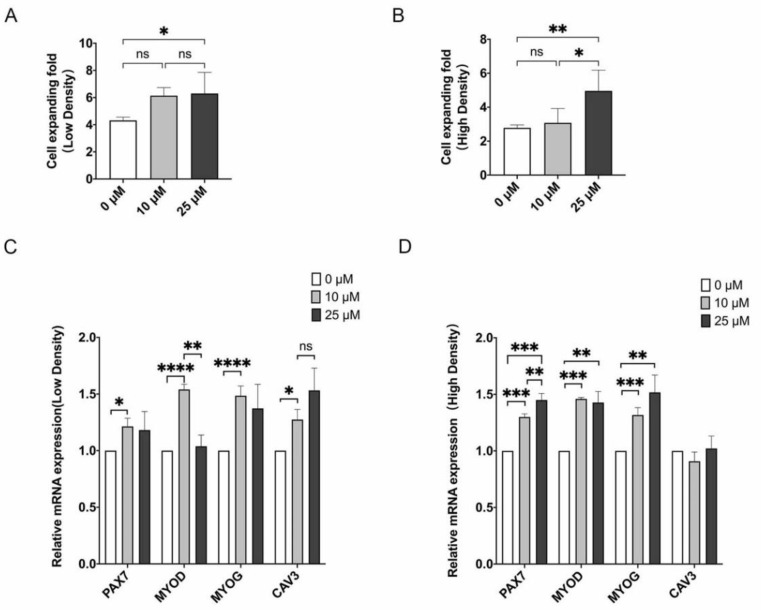
LPA promotes MuSCs proliferation and up-regulates myogenic genes. MuSCs were plated at low and high-density culture conditions. On the second day, cells were treated with 10 μM or 25 μM LPA and harvested after another 24 h. (**A**,**B**) The cell expansion ratio at low or high seeding densities with or without LPA treatment. (**C**,**D**) Expression by qPCR of *PAX7*, *MYOD*, *MYOG*, *CAV3* in MuSCs cultured with LPA under low or high-density conditions. Mean ± SD from three independent experiments. Significance was analyzed by ANOVA with Fisher’s protected least significant difference (PLSD). * *p* < 0.05, ** *p* < 0.01, *** *p* < 0.001, **** *p* < 0.0001, ns, not significant (*p* > 0.05).

**Figure 5 cells-10-03069-f005:**
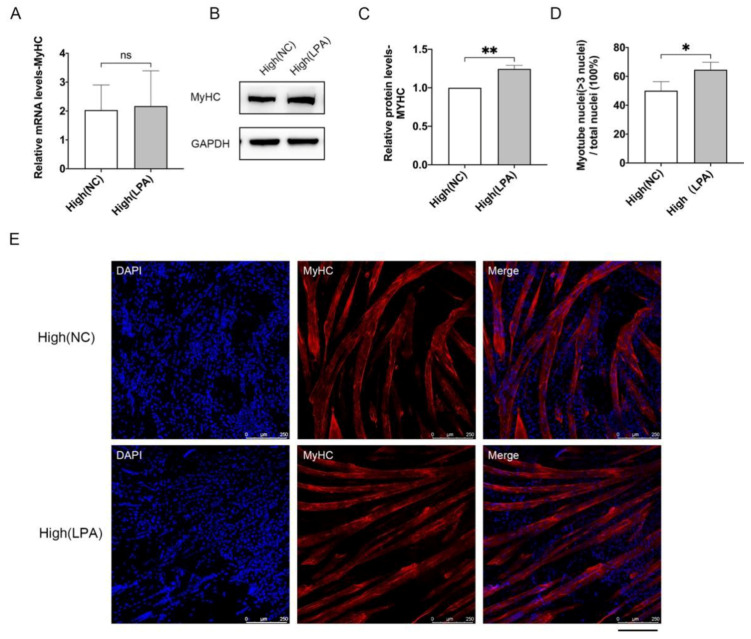
Differentiation capacity of MuSCs treated with or without LPA. MuSCs were plated at density of 2.2*10^4^/cm^2^ (high). On the second day, cells were treated with 25 μM LPA. After 24 h, cells were harvested and re-plated onto Matrigel-coated dishes to differentiate for 5 days. (**A**) Expression of *MYHC* by qPCR analysis. (**B**) MYHC protein levels by Western blot. (**C**) Western blot analysis of MYHC by Quantity One software. (**D**) Quantification of the percentage of myotubes with more than three nuclei indicated from (**E**). (**E**) Immunofluorescence staining of MYHC. Scale bar, 25 μM. Mean ± SD from three independent experiments. Significance was analyzed by Students’ *t*-test. * *p* < 0.05, ** *p* < 0.01, ns, not significant (*p* > 0.05).

**Figure 6 cells-10-03069-f006:**
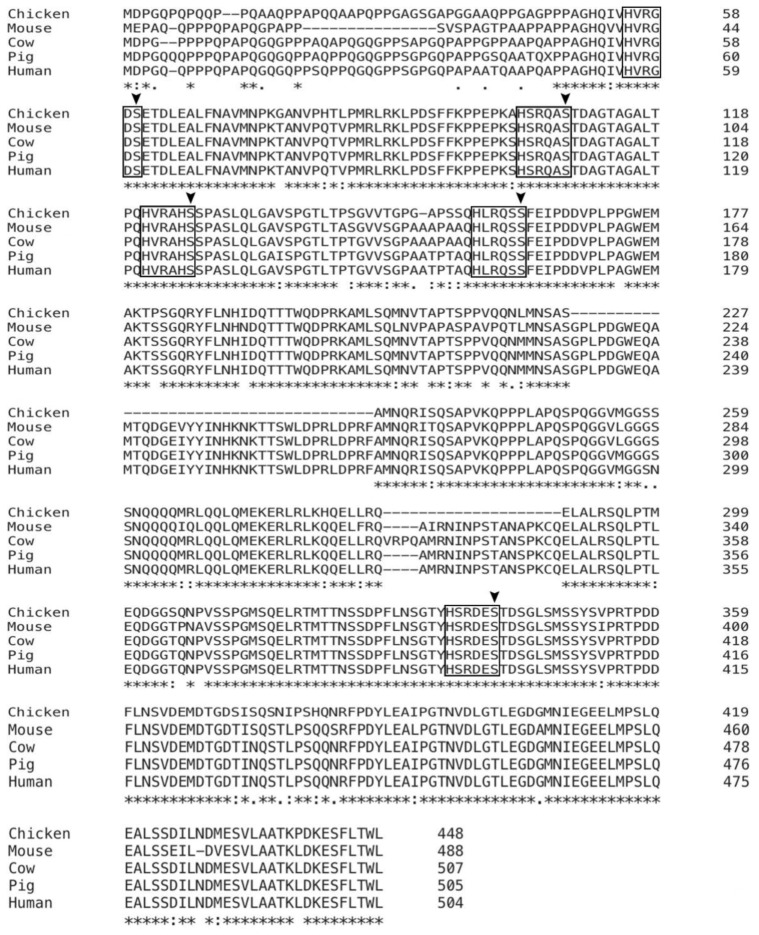
Alignment analysis of YAP proteins among different species. Schematic representation of the YAP protein sequences from pig and other species. The sequences enclosed by black frames indicate five conserved phosphorylation motifs HxRxxS (H: histidine, R: Arginine, S: serine) of YAP protein, and key serines are indicated by arrowheads.

**Figure 7 cells-10-03069-f007:**
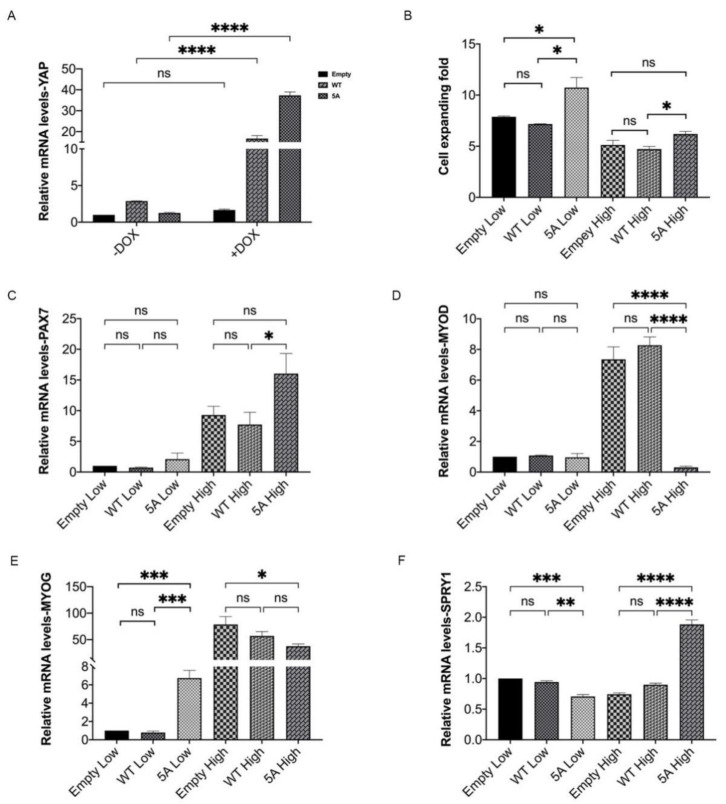
Overexpression of constitutively active YAP promotes cell proliferation and stemness maintenance of MuSCs. The 2 μg/mL Dox was used for 48 h to induce the expression of YAP before harvesting. (**A**) Expression of *YAP* induced by Dox in control MuSCs transfected with empty vectors and MuSCs transfected by wild-type (WT) YAP and phosphorylation mutational YAP (5A). (**B**) The cell expanding fold of MuSCs after Dox induction under low and high densities. (**C**–**F**) Expression of *PAX7*, *MYOD*, *MYOG,* and *SPRY1* under indicated conditions by qPCR analysis. Mean ± SD from three independent experiments. Significance was analyzed by two-way ANOVA for (**A**) and one-way ANOVA for (**B**–**F**). * *p* < 0.05, ** *p* < 0.01, *** *p* < 0.001, **** *p* < 0.0001, ns, not significant (*p* > 0.05).

**Table 1 cells-10-03069-t001:** Primers for qPCR analysis.

Genes	Primer Sequences (5′–3′)		Accession No.
*YAP*	Forward	TATCAACCAAAGCACCCTACC	XM_021062706.1
	Reverse	CTCCTCTCCTTCTATGTTCATTCC	
*PAX7*	Forward	GTGCCCTCAGTGAGTTCGATT	XM_021095458.1
	Reverse	TCCAGACGGTTCCCTTTGTC	
*MYOD*	Forward	GCTCCGCGACGTAGATTTGA	NM_001002824.1
	Reverse	GGAGTCGAAACACGGGTCAT	
*MYOG*	Forward	AACCCCACTTCTATGACGGG	NM_001012406.1
	Reverse	TTATCTTCCAGGGGCACTCG	
*MYHC*	Forward	CCGTGCTCCGTCTTCTTTCC	NM_001104951.2
	Reverse	CGCTCCTTCTCTGACTTGCG	
*SPRY1*	Forward	GCATAGACCTACCAGCCACC	NM_001267835.1
	Reverse	TCCGGATTGCCCTTTCAGAC	
*GAPDH*	Forward	GTCGGAGTGAACGGATTTGGC	NM_001206359.1
	Reverse	CTTGCCGTGGGTGGAATCAT	

## Data Availability

The data presented in this study are available on request from the corresponding authors.

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
