# Peer review of "YAP Promotes Cell Proliferation and Stemness Maintenance of Porcine Muscle Stem Cells under High-Density Condition"

_cells, 2021, doi:10.3390/cells10113069_

Round 1
Reviewer 1 Report
Muscle stem cell (MuSC) culture has a potential for producing meat in vitro. However, for that to happen, it would be highly beneficial if MuSC culture would not be limited by cell-cell inhibition. Liu et al. show here that porcine MuSC culture is inhibited by cell-cell contact and provide evidence that such inhibition is mediated by the Hippo-YAP pathway, which is known for fulfilling this role in other systems. Finally, they show that inhibiting the pathway by activating YAP can prevent cell-cell inhibition and yield higher MuSC growth, including at high densities. This is an interesting paper which could be appropriate for Cells, but it presents a number of issues that need to be fixed before publication.
Major points:
- It seems contradictory that the expression of both the MuSC gene PAX7 and the myogenic commitment gene MYOG are upregulated in MuSC grown under high-density conditions. Interestingly, expression of constitutively-active YAP is also increased in high-density cultures. Authors should discuss these results and propose possible explanations. Could expression levels of all genes more or less increase in contact-inhibited high-density cultures? Or is it simply the expression of the control gene GAPDH that decreases? As such, other house-keeping genes should be tested.
- All Western blots need to be replicated, quantified and analysed statistically. Moreover, the bands in some images look saturated (e.g. Fig. 5B) and are difficult to interpret.
- For Fig. 3B-C, could authors also show the non-phospho-YAP levels in both low- and high-density conditions, as this is the active and relevant form?
- For Fig. 3D, could the authors measure the fluorescence intensities for nuclear YAP in both low- and high-density conditions? And nucleo-cytoplasmic ratios?
- There are significant divergences between the values obtained for the cell expansion assay in control conditions in Fig. 1B (6-fold low vs 2-fold high), Fig. 4A-B (4-fold low vs 3-fold high) and Fig. 7B (8-fold empty low vs 5-fold empty high). As the results go in the same direction (i.e. expansion is always greater under low-density), this is not necessarily a problem in itself, but since the error bars within any one of the three experiments do not include the observed range seen across the three experiment, it is confusing. I would ask that authors repeat the experiment presented in Fig. 1B several times to get a sense of the real mean and of the intrinsic variability of the assay.
- The conclusion that “constitutively active YAP promotes cell proliferation and stemness of MuSC especially under high-density” is not convincing. In Fig. 7B, 5SA high is said to be significantly higher than empty high, but also not significantly different from WT high, which is even farther apart, and with less variability. How could that be? I would also recommend optimizing DOX concentration to find one that might work better. Perhaps too much over-expression (i.e. 40-fold) becomes detrimental?
- Differentiation capacity of MuSC expressing constitutive YAP should be assessed using the assay presented in Fig. 5D.
Minor Points:
- Primary literature referencing should be prioritized, not reviews.
- Replace YAP(5SA) by YAP(5A) throughout text and figures – this is a common practice.
- Potential pharmacological means to activate YAP should also be discussed in the last paragraph of the discussion.
- Finally, has anyone ever tasted cell cultured meat? Anything encouraging that could be referenced in the introduction?
Author Response
Thank you very much for reviewing our manuscript, and thank you for providing the constructive advice on helping improving the manuscript. We have revised the manuscript thoroughly according to the comments from you and other referees. We hope that you will be satisfied with our responses to the comments and the revisions for our manuscript.
For detailed responses point by point, please see the attachment.

Reviewer 2 Report
The manuscript intitled “YAP promotes cell proliferation and stemness maintenance of porcine muscle stem cells under high-density condition” aimed to investigate the roles of Hippo-YAP signaling with the objective of removing the cell contact inhibition and promoting the multiplication of pig muscle cells, in order to produce larger quantity of artificial meat.
The introduction provides information on the objectives and knowledge relating to the Hippo-Yap pathway which controls contact inhibition of cells. The experimental design is good, the methods well chosen to meet the scientific objective and the manuscript is particularly well written, detailed and clear. The results are clearly presented and very well discussed in a specific section. Thus in my opinion, the manuscript is very good.
I just have some minor comments:
Introduction
Line 31: I suggest to replace “Meat is the good source of high quality proteins” by “Meat is a good source of high quality proteins”
Materials and Methods
Line 89: Please specify if the piglet was sacrified or if the fresh muscle tissue came from a muscle biopsy. Please specify which skeletal muscle was used and its quantity.
Line 110 and below in the text. Check if 10^5 is correct for the journal (105 is more usual). If not correct, please change in the whole manuscript.
Line 195: “ “porcine” (not procine). However, the sentence could be removed as it is already mentionned in the M & M section (line 89)
Results
For the immunofluorescence images, did you make the controls by excluding the first antibody (but using the second one bind to the fluorescent dye) ? Did you have background ? please specify.
Discussion
Lines 377-378: “These results indicate increased, but not nucleic and functional YAP in MuSCs at high cell density ». This sentence is not clear to me. Please check and rephrase if necessary.
Author Response
Thank you very much for reviewing our manuscript and providing constructive suggestions. We have revised the manuscript thoroughly according to the comments from you and other referees. We hope that you will be satisfied with our responses to the comments and the revisions for our manuscript.
For detailed responses point by point, please see the attachment.

Round 2
Reviewer 1 Report
Authors have addressed most comments satisfactorily and as such, I would support publication. My only remaining concern is the relatively poor effect of constitutively active (CA) YAP transfection vs LPA. As author's suggest, this could be linked to a problem with their transfection technique. As a reader, we expect that the CA-YAP may promote expansion even better than the LPA, but it was clearly not the case: CA-YAP worked, but just enough to achieve significance. I would suggest to include a few sentences and a supplementary figure to mention the unexpected poor differentiation of transfected controls and samples, to let readers make up their own minds about this issue.